# Interaction of Conazole Pesticides Epoxiconazole and Prothioconazole with Human and Bovine Serum Albumin Studied Using Spectroscopic Methods and Molecular Modeling

**DOI:** 10.3390/ijms22041925

**Published:** 2021-02-15

**Authors:** Katarína Golianová, Samuel Havadej, Valéria Verebová, Jozef Uličný, Beáta Holečková, Jana Staničová

**Affiliations:** 1Faculty of Science, Pavol Jozef Šafárik University, Jesenná 5, 041 54 Košice, Slovakia; katagolianova@gmail.com (K.G.); s.havadej@gmail.com (S.H.); jozef.ulicny@upjs.sk (J.U.); 2University of Veterinary Medicine & Pharmacy, Komenského 73, 041 81 Košice, Slovakia; valeria.verebova@uvlf.sk (V.V.); beata.holeckova@uvlf.sk (B.H.); 3First Faculty of Medicine, Charles University, Kateřinská 1, 121 08 Prague, Czech Republic

**Keywords:** epoxiconazole, prothioconazole, interaction, serum albumins, molecular modeling

## Abstract

The interactions of epoxiconazole and prothioconazole with human serum albumin and bovine serum albumin were investigated using spectroscopic methods complemented with molecular modeling. Spectroscopic techniques showed the formation of pesticide/serum albumin complexes with the static type as the dominant mechanism. The association constants ranged from 3.80 × 10^4^–6.45 × 10^5^ L/mol depending on the pesticide molecule (epoxiconazole, prothioconazole) and albumin type (human or bovine serum albumin). The calculated thermodynamic parameters revealed that the binding of pesticides into serum albumin macromolecules mainly depended on hydrogen bonds and van der Waals interactions. Synchronous fluorescence spectroscopy and the competitive experiments method showed that pesticides bind to subdomain IIA, near tryptophan; in the case of bovine serum albumin also on the macromolecule surface. Concerning prothioconazole, we observed the existence of an additional binding site at the junction of domains I and III of serum albumin macromolecules. These observations were corroborated well by molecular modeling predictions. The conformation changes in secondary structure were characterized by circular dichroism, three-dimensional fluorescence, and UV/VIS absorption methods.

## 1. Introduction

Epoxiconazole (1-[[3-(2-chlorophenyl)-2-(4-fluorophenyl)oxiran-2-yl]methyl]-1,2,4-triazole, Figure 1a) is produced by BASF Corporation for protecting crops. Epoxiconazole is the active ingredient used in Opal 7.5 EC proposed for control of Black Sigatoka (*Mycosphaerella fijiensis*) and Yellow Sigatoka (*Mycosphaerella musicola*) in bananas and Coffee Rust (*Hamileia vastatrix*) in coffee [1]. Prothioconazole (2-[2-(1-Chlorocyclopropyl)-3-(2-chlorophenyl)-2-hydroxypro-pyl]-1,2-dihydro-3H-1,2,4-triazole-3-thione, Figure 1b) is the active ingredient used in Prosaro 250 EC for the control of *Pyriculari agrisea* and *Sclerotinia sclerotiorum* on corn and other commercial crops [2].

As azoles, epoxiconazole (EPX) and prothioconazole (PTC) actively stop the production of new fungi spores and inhibit the biosynthesis of ergosterol. Inhibition of lanosterol-C14-demethylase (CYP51) leads to accumulation of methylated sterols in the fungal membrane, thereby impairing its function. However, enzyme inhibition is not restricted to CYP51 (a fungal enzyme). Other enzymes inhibited by conazole fungicides include aromatase (CYP19), which converts androstenedione to estrogen, and 17-hydroxylase (CYP17), which converts progesterone into 17-hydroxy-progesterone, a precursor of cortisol and androsterone [2]. The enzyme 17-Hydroxylase also converts pregnenolone into 17-pregnenolone, a precursor of dihydroepiandrosterone, which is further metabolized to estrone and estradiol by aromatase. Thus, aromatase and 17-hydroxylase are key enzymes in steroidogenesis [3].

EPX is characterized by acute oral, dermal, and inhalation toxicity. It was administrated in a feed study to male and female mice, and both subsequently displayed significant increasing trends in liver adenomas, carcinomas and adenomas, and carcinomas combined [4]. Schneider et al. observed reduced estrogen levels after EPX treatment in rats [5]. The results obtained by Taxvig indicate that EPX is primarily fetotoxic, and secondly alters sex hormone levels in the dams, but not in the fetus during the developmental stages. The effects on reproductive development after perinatal exposure strongly indicate that one of the main mechanisms responsible is disturbance of key enzymes involved in steroid hormones synthesis [6]. EPX may also antagonize the androgen receptor and inhibit testosterone formation in vitro [7]. PTC mainly disrupts the endocrine system and has the potential to affect the human reproductive system [8]. Several studies have shown that PTC reduces cell proliferation (GH3 tumor cells) and inhibits the basal activity of the aryl hydrocarbon receptor (AhR) [9]. In vitro tests have shown that peripheral blood lymphocyte proliferation in cattle was almost completely inhibited within 48 h after exposure (30 µg/mL) to PTC [10].

Serum albumins (SAs) are the most abundant proteins in the blood system of a variety of organisms. Being the major macromolecules (60% of total protein, 3.5–5 g/dL) [11] contributing to osmotic blood pressure [12], they can also play a dominant role in drug distribution and efficacy in the body [13]. The two most studied serum proteins are human serum albumin (HSA) and bovine serum albumin (BSA). There are some differences between BSA and HSA, however. BSA has almost 88% sequence similarity with HSA, but their binding affinity for the same ligand is found to be invariably different [14,15]. They both consist of three homologous domains (I–III), each of which can be divided into two subdomains (A and B) [16,17]. BSA is a single-chain transporting protein consisting of 583 amino acids with 20 tyrosyl residues (Lys, Glu, Ala, Phe, Arg, Ile, Gn, His, Met, Asp, Ser, An, Tyr, Cys, Leu, Pro, Val, Thr, Ileu, Gly [18]) and two tryptophanes located in positions 134 and 213 [19]. HSA consists of 585 amino acids with 17 tyrosyl residues (Lys, His, Arg, Asp, Thr, Ser, Glu, Pro, Gly, Ala, Cys, Val, Met, Ile, Leu, Tyr, Phe [20]) and Trp located in position 214. HSA has tryptophan (Trp214) in subdomain IIA, whereas BSA has two tryptophan moieties (Trp134 and Trp213) located in subdomains IA and IIA respectively [19,21]. Comparison of the amino acid sequence of both the serum proteins reveals that 15 amino acid residues in subdomain IIA of HSA are replaced in the bovine variety [22].

HSA has extraordinary binding properties. It can bind under the physiological conditions not only fatty acids, peptides, and proteins, but also low molecular weight endogenous and exogenous molecules [23,24]. HSA binds medium and long saturated fatty acid molecules (C10–C18) and long chain unsaturated fatty acid molecules (arachidonic and oleic acid) up to seven binding sites (FA1–FA7). They are located inside subdomains (FA1, FA4, FA5, and FA7), at the interface of domains (FA2 and FA3) and at the interface of two subdomains belonging to one domain (FA6) (Figure 2a). HSA has a high affinity site for binding metals to N-ends of the chain [11], where bind mainly Zn(II), Cd(II), Cu(II) and Ni(II) [25]. Domains II and III contain two primary drug binding sites, known as Sudlow’s site I and II (Figure 2a) [26]. Sudlow´s site I is the primary binding site for warfarin, often called warfarin binding sites, but molecules such as phenylbutazone and azapropazone also bind there. It is located in cavity of subdomain IIA, which is for the most part nonpolar. An important part of Sudlow’s site I is the existence of only one Trp present in HSA (Trp214), as its indole ring and its rotation allow by a certain ligand nesting into the Sudlow’s site and creating the stacking interactions. Molecules that specifically bind into the Sudlow’s site I are bulky negatively charged heterocyclic molecules located in the middle of them. Sudlow’s site II is the primary binding for ibuprofen and diazepam. Compared to Sudlow’s site I, it is smaller and contains only one binding region. While Sudlow’s site I is surrounded by subdomains IIB and IIIA, Sudlow’s site II is surrounded by subdomains IIIB and IIIA, which different rotation provides direct contact of Sudlow’s site II with the solvent. Sudlow’s site II binding molecules are predominantly aromatic carboxylic acids, which have negative charge on the alpha carbon [27,28,29].

BSA has several binding sites, of which between the most important are sites I and II located in the hydrophobic cavities of subdomains IIA and IIIA [31]. Markers of these binding sites include warfarin, phenylbutazone, dansylamide, iodipamide (binding site I), and ibuprofen, flufenamic acid, and diazepam (binding site II) (Figure 2b). Unique binding properties are manifested in binding with nucleic acids, proteins, and coordination compounds [32,33,34]. BSA also has the ability to bind large amounts of ions (Cu^2+^, Ni^2+^, Zn^2+^, Co^2+^, Pt^2+^) [35,36,37]. Some degree of binding activity was also observed for metal complexes due to weak binding interactions between metals and the tryptophan residue Trp134, which is located on the surface (it is more available than Trp213) [38].

The binding of pesticides to HSA and BSA has seldom been investigated. Silva et al. studied the binding of methyl parathion (organophosphorous pesticide) to HSA and BSA in order to establish whether methyl parathion exhibited a high affinity to HSA and BSA [7]. Wang et al. and Yan et al. studied the interaction between imidacloprid (neonicotinoid pesticide) and SAs (HSA and BSA) [39,40]. Cui et al. studied the binding of chlorpyrifos (organophosphate pesticide) and cypermethrin (syntetic pyrethroid) to blood proteins [21]. Binding of pesticides to plasma proteins has toxicological importance, and significantly affects the distribution and excretion of pesticides. Our study is expected to provide an important insight into the interactions of the transport proteins HSA and BSA with fungicides EPX and PTC, representatives of the conazoles group. We studied the binding of fungicides with HSA and BSA using spectroscopic methods including fluorescence, absorbance, and circular dichroism complemented with molecular modeling.

## 2. Results and Discussion

### 2.1. Fluorescence Spectra of Conazoles–SA System

To investigate the interactions between pesticides and biomacromolecules, we used the very effective method of fluorescence spectroscopy. The presence of aromatic acids at the binding sites of HSA and BSA biomacromolecules led us to use 280 nm as the excitation wavelength in our measurements of fluorescence quenching. We used the excitation wavelength 295 nm to separate the fluorescence spectra of tyrosine (Tyr) and tryptophan (Trp). The fluorescence emission spectra of both HSA and BSA upon the addition of EPX measured at 25 °C are shown in Appendix A and Appendix A respectively (see Appendix A). It can be seen that the fluorescence intensities of SAs decreased regularly with increasing concentrations of EPX. No shifts in maximum emission wavelength of HSA and BSA were observed, suggesting that small molecules like pesticides are likely to bind into SAs via the hydrophobic region located inside the protein [41]. Upon addition of PTC it could be seen that the intensity of fluorescence of HSA first increased (to ratio 2/1 PTC/HSA) and then regularly decreased (Appendix A, see Appendix A). On the other hand, the intensity of fluorescence of BSA decreased regularly (Figure 3). We assume, that the increasing fluorescence intensity of Trp could be caused by the existence of two independent binding sites. Based on the fact that there are two binding sites in the PTC/HSA complex, we can deduce that PTC binds first to a non-specific site and causes a slight structural opening (which leads to increasing fluorescence intensity). After the structural changes, Trp becomes more available for PTC binding. The maximum emission wavelength of HSA upon the addition of PTC was almost unchanged. The maximum intensity of BSA upon the addition of PTC was very slightly shifted to the left (3 nm) (Figure 3, inset), which means that the polarity around Trp was very slightly changed.

Fluorescence quenching can be induced by processes such as excited state reactions, molecule rearrangements, energy transfers, and ground state complex formation. Quenching mechanisms are generally classified into dynamic quenching and static quenching, which can be distinguished by their different dependence on viscosity, lifetime measurement, and temperature [42].

Dynamic quenching depends mainly on diffusion, so higher temperatures result in larger diffusion coefficients and quenching constants are expected to increase with increasing temperature. On the other hand, since rising temperature tends to decrease the stability of complexes, the static quenching constant values are expected to be lower [42,43].

The quenching mechanism can be described by the Stern–Volmer Equation (1) [44]:(1)$$\frac{F_{0}}{F}=1+\;K_{q}\tau_{0}=\;1+\;K_{SV}\left[Q\right]$$
where *F*_0_ and *F* are the fluorescence intensities of protein in the absence and in the presence of quencher respectively. *K_q_* and *K_SV_* are the biomolecule quenching rate constant and the Stern–Volmer constant respectively. $\tau_{0}$ is the average lifetime of the biomolecule without quencher and [*Q*] is the quencher concentration. The inner filter effect was eliminated by using a very low concentration of HSA, which gives absorbance of less than 0.1 at excitation wavelength 295 nm. The plots of *F*_0_/*F* versus [*Q*] at three different temperatures are shown in Figure 4. The plots indicate a single type of quenching phenomenon. The slope of the quenching curves decreases with increasing temperature, implying that the quenching mechanisms of EPX/HSA; EPX/BSA, and PTC/HSA; PTC/BSA were initiated by complex formation rather than dynamic collision.

### 2.2. Determination of Binding Parameters

For the static quenching interaction, binding parameters such as additional binding sites in the biomolecule (*n*) and the association constant of complex (*K_A_*) could be determined using the Hill Equation (2) [12,22,41]:(2)$$\log\left[\frac{F_{0}-F}{F}\right]=\log K_{A}+n\log\left[Q\right]$$
where *F*_0_, *F* are the fluorescence values in the absence and presence of quencher and [*Q*] is quencher concentration. *K_A_* and n can be expressed as the slope and intercept *of double logarithm regression* of log [(*F*_0_ − *F*)/*F*] vs. log [*Q*] based on Equation (2). Hill plots for PTC/BSA interaction can be seen in Figure 5 and for EPX/HSA, EPX/BSA, and PTC/HSA in Appendix A respectively (see Appendix A). The corresponding data for binding parameters at different values are summarized in Table 1.

As can be seen in Table 1, the association constants of EPX/HSA; EPX/BSA, and PTC/HSA; PTC/BSA complexes suggest strong binding affinity. The *n* values of EPX/HSA; EPX/BSA, and PTC/BSA complexes are approximately equal to 1, suggesting that there is no additional binding site given by cooperation with regular binding site in both HSA and BSA. For the PTC/HSA complex, the n value is also close to 1 which means that the other additional binding site is not in the vicinity of Trp. HSA has a large number of binding sites. For small ligands, these are mainly Sudlow site I (in domain IIA) and Sudlow site II (in domain IIIA), or small ligands can additionally bind along the protein cleft between domains IB–IIIA [45].

### 2.3. Thermodynamic Parameters

The interaction forces between biomacromolecules and small molecules can include the van der Waals force, hydrogen bond, electrostatic force, and hydrophobic interaction force. Thermodynamic parameters such as Gibbs free energy change (Δ*G*), enthalpy change (Δ*H*), and entropy change (Δ*S*) can account for the main forces involved in the binding process. The values of listed thermodynamic parameters can be calculated from the van’t Hoff Equation (3):(3)$$\ln K_{A}=\frac{-∆H}{RT}+\;\frac{∆S}{R}$$
where *K_A_* is the association constant at temperature *T* and *R* is the universal gas constant (*R* = 8314 J/K.mol). The binding studies were recorded at 298, 303, and 310 K. By plotting ln*K_A_* vs. *1/T* it was possible to determine the values of Δ*H* and Δ*S*. Thus, Δ*G* could be calculated using Equation (4):(4)$$∆G=\;∆H-T∆S=\;-RT\ln K_{A}$$

It has been defined, in previous studies [46,47] that the magnitudes and signs of thermodynamic parameters correlate with various individual kinds of interaction. For negative values of Δ*H* and Δ*S*, van der Waals forces and hydrogen bond formation are suggested as more important, whereas for a negative value of Δ*H* and positive value of Δ*S* the hydrophobic interaction plays a major role. The calculated values of thermodynamic parameters are summarized here in Table 1. The negative values of Δ*G* indicate that these binding processes are spontaneous. The negative values of Δ*H* and Δ*S* suggest that the hydrogen bond and van der Waals forces play a major role in the interaction [48]. This claim can be supported by the fact, that nitrogen atoms, located on the triazole ring, can easily form hydrogen bonds with some amino acid residues such as Trp, Arg, and Lys. Similarly, the OH group in PTC is the hydrogen bond donor.

### 2.4. UV/Vis Spectroscopy

UV/Vis spectroscopy is an effective method for exploring the structural changes in protein and ligand/protein complex formation. Figure 6 and Appendix A show the absorption spectra of EPX/HSA, PTC/HSA complexes and EPX/BSA, PTC/BSA complexes respectively.

The absorption spectra of BSA and HSA show a strong peak at 207 nm (peak 1) characterized by *n*→π* transfers in the polypeptide structure of proteins. Another peak at 280 nm (peak 2) is characterized by π→π* transfers in the amino acids in proteins. As can be seen in Figure 6 and Appendix A, the absorption intensities of BSA and HSA decreased with the addition of EPX and PTC, indicating that pesticide/BSA and pesticide/HSA complexes were formed. After EPX binding to BSA and HSA, the maximum peak 1 position shifted towards the higher wavelength region (8 nm), but after PTC binding only a very slight red shift (2 nm) was observed in the maximum peak 1 position for BSA and HSA. This indicates a change in polarity of the peptide strand of BSA and HSA biomolecules, and hence a change in hydrophobicity [49,50]. The binding of EPX and PTC to BSA and HSA might lead to change in protein conformation [51]. This also indicates that the peptide strands of albumin molecules extended further upon the addition of EPX and PTC to BSA and HSA [49]. No changes in the wavelength shift of peak 2 were observed.

### 2.5. Synchronous Fluorescence Spectroscopy Studies

Synchronous fluorescence spectra can give information about the molecular environment in the vicinity of a chromophore (Trp, Tyr) and spectroscopy includes simultaneous scanning of the excitation and emission monochromators while maintaining a constant wavelength interval (Δλ) between them [52]. When Δλ were settled at 15 nm and 60 nm, the spectra provided individual information for amino acids residues Tyr and Trp in the protein respectively [53]. The synchronous fluorescence spectra for Trp and Tyr amino acid residues on HSA and BSA at various concentrations of EPX and PTC are shown in Figure 7 and Appendix A. Figure 7 shows strong fluorescence quenching of Trp residues and no fluorescence quenching of Tyr in the PTC/BSA complex. The decrease in intensity of fluorescence for Trp was stronger than for Tyr residues, indicating that Trp residues contributed more to the quenching of the intrinsic fluorescence. In PTC/BSA we observed a very slight shift in maximum emission wavelength (3 nm), suggesting that the microenvironment around Trp was changed. In Appendix A as the concentration of EPX and PTC increased regularly, the intensity of fluorescence of Trp residues in HSA and BSA decreased. We also observed slight fluorescence quenching of Tyr residues after PTC and EPX binding to HSA, indicated that the binding site is in the vicinity of both Tyr and Trp. The unchanged maximum emission wavelength suggested that polarity around Trp and Tyr remained constant [54]. These results are consistent with those of the site marker experiments.

### 2.6. Site Marker Competitive Experiments

Site marker competitive experiments were performed to elucidate the binding sites of EPX and PTC to BSA and HSA. The measurements were carried out in the presence of two nonsteroidal anti-inflammatory drugs ketoprofen (KPF) and ibuprofen (IBF). KPF and IBF have been identified as stereotypical ligands for Sudlow’s sites I and II (subdomain IIA and IIIA), respectively [55]. For the analysis we used the method originating from Sudlow et al. [56], which dealt with probe (KPF, IBF) displacement and can be expressed by the following Equation (5):(5)$$Probe\;displacement=\;F_{2}/F_{1}$$
where *F*_2_ and *F*_1_ represent the intensities of fluorescence of EPX/BSA; EPX/HSA and PTC/BSA; PTC/HSA systems in the presence and absence of the probe respectively. It can be seen (Appendix A) that the increasing concentration of KPF led to decreasing intensity of fluorescence in the pesticide/BSA; pesticide/HSA complex. On the other hand, the increasing concentration of IBF had a little impact on intensity of fluorescence in the pesticide/BSA; pesticide/HSA complex. The displacement of EPX bound to BSA and HSA and PTC bound to HSA after addition of KPF indicated that site I (subdomain IIA) was the probable binding site of EPX to BSA and HSA and PTC to HSA. Another result was observed for the interaction of PTC with BSA (Figure 8). The increasing concentration of KPF and IBF had only slight impact on intensity of fluorescence in the PTC/BSA complex, which suggests that site I and site II were not probable binding sites of PTC to BSA. In this case we can consider the binding of PTC to the surface of the BSA molecule. As it will be shown below in the text, our further results confirm this proposition and this binding site is located on domain split in the albumin molecule (Figure 13).

### 2.7. Three-Dimensional Fluorescence Spectroscopy

Three-dimensional fluorescence spectra can provide more detailed information about conformational change in protein [57]. If there is a shift in the emission and excitation wavelength around the fluorescence peak, or a new peak appears, or the existing peak disappears, this can be an important indicator suggesting conformational changes in BSA and HSA. The micro-environmental and conformational changes in BSA and HSA were investigated by comparing their spectral changes in the presence and absence of EPX and PTC (Figure 9 and Appendix A). Peak a (λ_ex_ = λ_em_) and peak b (2λ_ex_ = λ_em_) are Rayleigh scattering peaks of first and second order. They arise from elastic light scattering on particles smaller than the light wavelength. The intensity of both scattering peaks increases with the addition of EPX and PTC resulting in an increase in the diameter of the molecule and thus greater scattering [58,59]. Peak 1 mainly reveals the spectral characteristics of Trp and Tyr residues on proteins, and the other peak 2 shows the behavior of the polypeptide backbone structure in BSA and HSA.

In Figure 9 and Appendix A it can be seen that the fluorescence intensity of peaks 1 and 2 decreased with the addition of EPX and PTC. Peak 1 represents the fluorescence contribution of Tyr and Trp and peak 2 represents the spectral properties of the polypeptide structural chain; its intensity correlated with the secondary structure of the protein [58]. We can observe a decreasing tendency of peak 1 after adding individual conazoles to SA solutions (Figure 9). This decrease indicates the binding of conazoles near the Tyr or Trp of the individual proteins, which is consistent with fluorescence measurements. We can also observe a decrease in peak 2, which indicates changes in the peptide structure. Its declining trend after conazoles binding suggests that there is a slight destabilization of individual proteins and a slight unfolding of the polypeptide chain leading to conformational changes [58]. This result correlates with absorption and CD spectroscopy measurements. No shifts in maximum emission wavelength were observed. This suggests that the conformation of BSA and HSA altered and EPX and PTC were located in the vicinity of Trp or Tyr residues on BSA and HSA.

### 2.8. Circular Dichroism

Using the previous techniques, we found some conformation alteration in BSA and HSA. To collect more information on the binding of pesticides to BSA and HSA we used the circular dichroism method. HSA and BSA present two negative bands in the ultraviolet region at 208 nm and 222 nm, characteristic for the α-helical structure [60,61]. Two negative peaks contribute to π→π* transfer (208 nm) and *n*→π* transfer (222 nm) for the peptide bond in the α-helix [62]. The CD measurements performed in the absence and presence of different concentrations of EPX and PTC revealed that the binding of pesticides to BSA and HSA caused a slight decrease in both bands (Figure 10 and Appendix A).

According to the following equation, the CD results were expressed in terms of mean residue ellipticity (*MRE*) (6):(6)$$MRE=\;\frac{Observed\;CD\;\left(mdeg\right)}{10cnl}$$
where *c* is the concentration of BSA and HSA, *n* is the number of amino acid residues of BSA and HSA and *l* is the path length.

The *α-helix* contents of free BSA, and HSA and pesticides/BSA; pesticide/HSA complexes were calculated from *MRE* values at 208 nm using the following Equation (7) [62]:(7)$$\alpha -helix\;\left(\%\right)=\;\frac{-MRE_{208\mathrm{nm}}-4000}{33000-4000}\;\times \;100$$

Free HSA and BSA contain 50.32% and 57.20% α-helical structures respectively. In the presence of EPX and PTC the content of α-helical structures gradually decreased. We used the web server K2D3 [63] to verify the content of α-helix and calculate β-strand of the protein from its circular dichroism spectrum. Using this program, we found that free HSA and BSA contained 12.69% and 11.69% β-strand respectively. By adding EPX and PTC to the solution, we noticed a gradual increase in β-structures. The increasing of β-strands was more modest than the decrease in α-helixes. We also applied ethanol to probe its effect on the secondary structures in HSA and BSA. The effect of ethanol was less than 1%, so we can state that EPX and PTC cause slight changes in the secondary structures of BSA and HSA.

These results show that pesticides interact with the amino acid residues of the polypeptide backbone in BSA and HSA, disrupt the hydrogen bonds [64] and evoke protein destabilization [59]. The CD spectra in the presence and absence of EPX and PTC have similar shapes, with no changes in peak positions. This indicates that the α-helical structure is still dominant. The results are in good agreement with those obtained from three-dimensional fluorescence spectroscopy.

### 2.9. Molecular Docking

The crystal structures of HSA (variant 1AO6) and BSA (variant 4FS5) were used as the target for docking. We used Autodock Vina for simulation which provides good accuracy for rigid docking simulation but its results may be inaccurate for docking with significant conformation change. But rigid docking simulation for small conformational change are still used in literature [12,65,66,67].

In the first iteration we did blind docking simulation which showed that the greatest binding affinity for EPX binding in HSA and BSA were at the Sudlow site I <=> FA7. Other locations (Sudlow site II <=> FA6) and surface sites) were also noted but the binding energy was at least 10% (Sudlow site II) or smaller (surface sites ranging in 20–50% less) than for the Sudlow site I. For this reason we reduced the further simulation space to 30 × 30 × 30 Å centered on the warfarin binding site.

Figure 11 and Figure 12 display the best binding modes of EPX to HSA; BSA, and PTC to HSA, BSA, respectively.

EPX in HSA was located deep in the hydrophobic cavity with binding energy −33.48 kJ/mol and surrounded by the following amino acid residues: Tyr150, Lys199, Trp214, Arg213, Leu219, Arg222, Phe223, Leu226, Leu260, Ile290, and Ala291 (Figure 11a). EPX was located in the vicinity of Tyr and Trp (nearer to Trp), which is in good agreement with the results from synchronous spectroscopy. We also observed the process of π-cation interaction between the triazole ring of EPX and hydrogen atom of Lys199. In BSA, EPX was located in the hydrophobic cavity with binding energy −34.33 kJ/mol. The amino acids surrounding this conformation were: Arg198, His211, Trp213, Arg217, Lys221, Phe222, Arg256, Ser286, Ile288, and Ala290. EPX binds to BSA via the π-cation interaction between the triazole ring of EPX with carbon atom Arg198 (Figure 11b). These results of our docking studies agree well with the ones from thermodynamic analysis (Table 1), showing that EPX can bind to HSA and BSA in subdomain IIA, and that they share a similar binding mode.

PTC in HSA was located deep in the hydrophobic cavity with binding energy −27.63 kJ/mol, surrounded by the following amino acid residues: Lys195, Trp214, Arg222, Lys436, Asp451, Tyr452, and Val455. PTC was located in the vicinity of Tyr and Trp (nearer to Trp), which is in good agreement with the results from synchronous spectroscopy. In Figure 12a we can also identify existence of π-cation interaction between the carbon atom of benzene in PTC and hydrogen atom in Lys436. The second binding site for PTC in HSA was located at the junction of domains I and III (Figure 13). PTC located in the protein cleft of albumin was stabilized by the hydrogen bond between the nitrogen atom of the triazole ring in PTC and hydrogen atom in Arg186 with binding energy −29.30 kJ/mol.

Regarding the docking of PTC to BSA, we observed rather different results. The docking results here need to be highlighted as the conformation change is bigger than by other samples and the approximation of small confirmation change might be strained and a more detailed docking simulation should be done in the future.

In BSA (Figure 12b) the most stable position of PTC was docked to the protein surface with binding energy −27.46 kJ/mol. The amino acids surrounding this conformation were: Lys20, Leu24, Val40, Val43, Lys132, Trp134, and Gly135. PTC binds to BSA via the π-π interaction between the benzene ring in PTC and Trp134. Direct binding of PTC to Trp caused change in the polarity of Trp, which agreed with our results from synchronous fluorescence spectroscopy, where a blue shift in maximum emission wavelength was observed. Surface binding was also corroborated by the results from our site marker competitive experiments. Binding PTC to the protein cleft of BSA was stabilized via the π-cation interaction between PTC and His145 with binding energy −28.46 kJ/mol. These docking study results agree well with the ones from our thermodynamic analysis (Table 1) and confirm that PTC can bind to HSA and to BSA with different binding modes.

## 3. Materials and Methods

### 3.1. Chemicals and Reagents

EPX (CAS Number 133855-98-8) and PTC (CAS Number 178928-70-6) with 99% purity were purchased from Sigma Aldrich (Darmstadt, Germany). The pesticides were dissolved in spectroscopic grade 100% ethanol at concentration 10^−3^ mol/L. HSA and BSA (fatty acid free, globulin free, purity no less than 99%) were also obtained from Sigma Aldrich (Darmstadt, Germany) and were used without further purification. Stock solutions of HSA and BSA (concentration 5 × 10^−4^ mol/L) were prepared in Tris-Cl^−^ (0.05 mol/L Tris + 0.1 mol/L NaCl) buffer, pH 7.4 and conserved in the dark at 4 °C. Ketoprofen and ibuprofen from Sigma Aldrich (Darmstadt, Germany), with purity no less than 98%, were dissolved in 100% ethanol at concentration 10^−3^ mol/L. Phosphate buffer (0.02 mol/L, pH 7.4) was used for CD spectroscopic measurements to block the influence of chloride ions. Tris(hydroxymethyl)aminomethane, NaCl, HCl, and other reagents were all of high purity. The buffers were prepared in triple-distilled water. All experiments were performed three times independently, and the presented data were summarized as mean values with related standard deviations.

### 3.2. Fluorescence Measurements

Fluorescence measurements were performed on a SHIMADZU RF 5301 PC spectrofluorimeter (Kyoto, Japan) in a 1 cm quartz cuvette. Excitation wavelength was set at 295 nm and fluorescence was collected from 300 to 500 nm using 5 nm/5 nm slits (HSA) and 3 nm/3 nm slits (BSA). Fixed concentrations of HSA and BSA (2 × 10^−6^ mol/L, 2.5 mL) with various concentrations of EPX and PTC were added to the sample cell by means of titration (to give a final volume 80 μL). After each titration the sample was stabilized for 10 min. Measurements in all experiments were taken at three different temperatures (25, 30, 37 °C).

Synchronous fluorescence spectra were generated at room temperature. The scanning intervals between excitation and emission wavelengths were fixed at 15 and 60 nm respectively. Emission spectra were recorded in the range 200–400 nm.

Site marker competitive experiments were carried out at room temperature. The concentrations of EPX, PTC and HSA, BSA were fixed at 2 × 10^−6^ mol/L. Ketoprofen and ibuprofen were then gradually added into the EPX/HSA; EPX/BSA and PTC/HSA; PTC/BSA complexes. The fluorescence quenching data for the four systems were recorded over the range 300 nm to 500 nm using slits 5 nm/5 nm (HSA) and 3 nm/3 nm (BSA) and an excitation wavelength of 295 nm.

Three-dimensional fluorescence spectra were also recorded at room temperature. The excitation wavelength was set at 220 nm with increments of 5 nm, and emission spectra were recorded in the range 250 nm to 500 nm.

The different values of HSA and BSA concentrations were adjusted to the optimal conditions of the measurement methods used.

### 3.3. UV/Vis Measurements

UV/Vis measurements were performed on a Carry 60 UV—Vis spectrophotometer (Agilent, Santa Clara, CA, USA) in a 1.0 cm quartz cuvette at room temperature. Concentrations of HSA and BSA (2 × 10^−6^ mol/L, 2.5 mL) were fixed while the pesticide concentrations were varied from 2 × 10^−6^ mol/L to 32 × 10^−6^ mol/L (to give a final volume 80 µL). Equal concentrations of EPX and PTC were added to the reference cell simultaneously. The UV/Vis spectra were collected in the range 200 to 350 nm.

### 3.4. Circular Dichroism (CD) Studies

CD spectroscopy measurements were performed on a Jasco J-815 CD spectrometer (Jasco, Easton, MD, USA) in a 0.1 cm quartz cell at room temperature with constant nitrogen flush. The CD spectra of HSA and BSA in the presence of EPX and PTC were recorded in the range 190 nm to 270 nm with scan rate 50 nm/min. Three scans were collected for each spectrum, taking the average as the final data. EPX/HSA; EPX/BSA and PTC/HSA; PTC/BSA complexes were prepared by means of titration of EPX and PTC into HSA and BSA (concentration 3 × 10^−6^ mol/L) to achieve final concentrations of EPX and PTC from 3×10^−6^ mol/L to 1.5 × 10^−5^ mol/L.

### 3.5. Molecular Docking

We used HSA and BSA crystallographic data from the Brookhaven Protein Data Bank. The species selected were 1AO6 (HSA) and 4F5S (BSA) [68], as these describe the proteins in their unbound state with sufficient precision. The molecular structures of EPX and PTC were taken from the PubChem database. For docking we used AutoDockVina [69]. Based on experimental results we believe that the rigid docking that is supported in AutoDockVina provides sufficiently good results to reality. Blind docking was first performed for initial interaction site targeting with the result of most energetically stable results being in the vicinity of the warfarin binding site with other binding energy falling by at least 10% or more. In the investigation we also included other binding sites which are mentioned in the literature [70]. Our main focus was thus on investigating EPX and PTC activity at the warfarin binding site. The 2D representations were done as described in [71].

## 4. Conclusions

The present paper focuses on results obtained from the study of EPX and PTC with blood proteins, represented by HSA and BSA. Investigation of the interaction of conazole fungicides with serum albumins is essential for the determination of their function in biological systems. In this study we investigated the binding of conazole fungicides with serum albumins using spectroscopic methods complemented with molecular modeling. EPX and PTC were found to interact with BSA and HSA in vitro under simulated physiological conditions. The albumin fluorescence quenching by these fungicides was attributed to a static mechanism. The binding and thermodynamic parameters confirmed structurally-related binding modes, which is in good agreement with the molecular modeling results. The displacement of probes from BSA and HSA after addition of the fungicides indicated that Sudlow site I (subdomain IIA) was their probable binding site to BSA and HSA with the exception of the PTC/BSA complex, where we suggest that PTC binds onto the surface of BSA. In PTC/HSA and PTC/BSA complexes we observed another more probable binding site due to conformal changes, giving better access for PTC to subdomain IIA. The binding of fungicides to BSA and HSA leads to slight conformational changes in albumins, as corroborated by our three-dimensional fluorescence and CD measurement outcomes. Our results are also in good agreement with those presented in other studies [42,51,52] where slight binding interactions and small changes in the secondary structure of BSA and HSA after binding with triazole fungicides were observed.

## Figures and Tables

**Figure 1 ijms-22-01925-f001:**
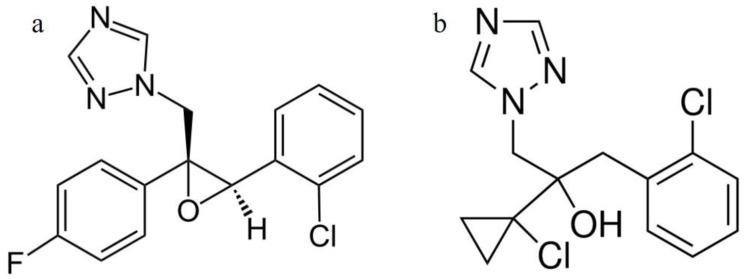
Chemical structures of Eepoxiconazole (EPX) (**a**) and prothioconazole (PTC) (**b**).

**Figure 2 ijms-22-01925-f002:**
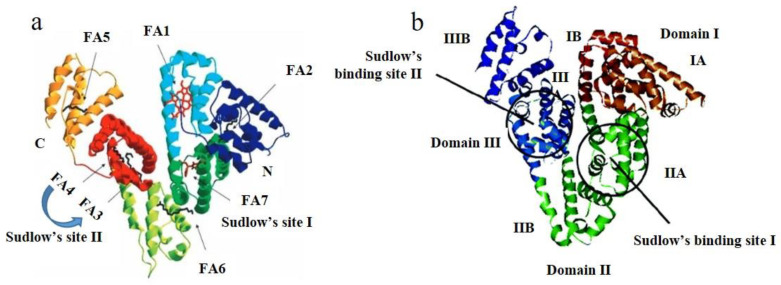
Structure and binding sites of human serum albumin (HSA) (**a**) (edited by [23]) and bovine serum albumin (BSA) (**b**) (edited by [30]).

**Figure 3 ijms-22-01925-f003:**
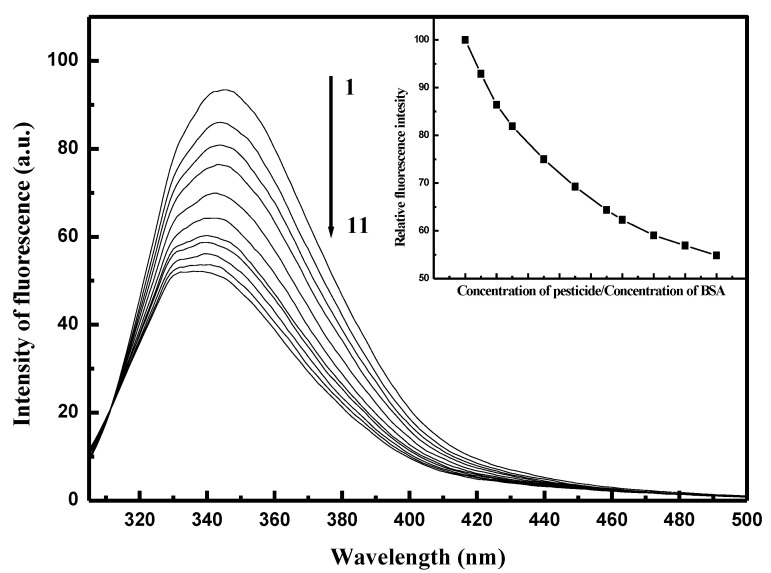
Quenching of bovine serum albumin (BSA) fluorescence after prothioconizole (PTC) binding. c (BSA) = 2 × 10^−6^ mol/L; c (PTC) = 2 × 10^−6^ mol/L–32 × 10^−6^ mol/L (1–11). Inset: Decrease in BSA fluorescence in the presence of PTC.

**Figure 4 ijms-22-01925-f004:**
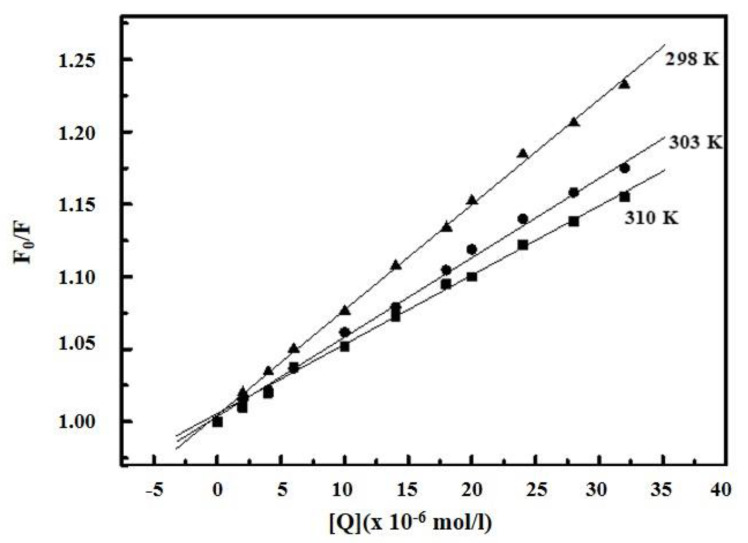
Stern–Volmer plots for prothioconazole (PTC) binding with bovine serum albumin (BSA) at three different temperatures.

**Figure 5 ijms-22-01925-f005:**
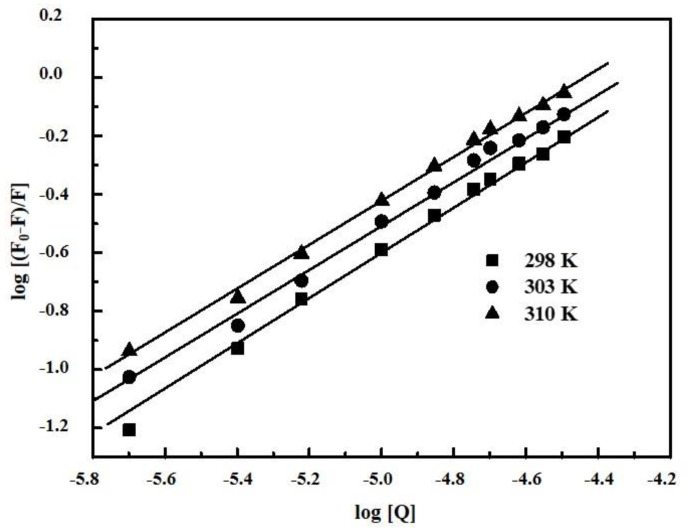
Hill plot for prothioconazole/bovine serum albumin (PTC/BSA) interaction. *T* = 298 K, 303 K and 310 K.

**Figure 6 ijms-22-01925-f006:**
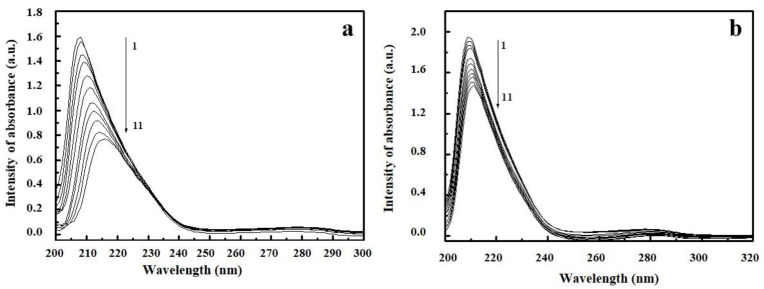
UV/Vis absorption spectra of human serum albumin (HSA) in the presence of epoxiconazole (EPX) (**a**) and prothioconazole (PTC) (**b**). c (HSA) = 2 × 10^−6^ mol/L; c (EPX, PTC) = 2 × 10^−6^ mol/L–32 × 10^−6^ mol/L (1–11).

**Figure 7 ijms-22-01925-f007:**
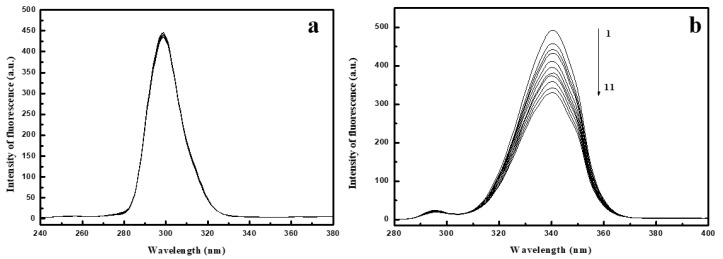
Synchronous spectra on Tyr (**a**) and Trp (**b**) of bovine serum albumin (BSA) in the presence of prothioconazole (PTC). c (BSA) = 2 × 10^−6^ mol/L; c (PTC) = 2 × 10^−6^ mol/L − 32 × 10^−6^ mol/L (1–11).

**Figure 8 ijms-22-01925-f008:**
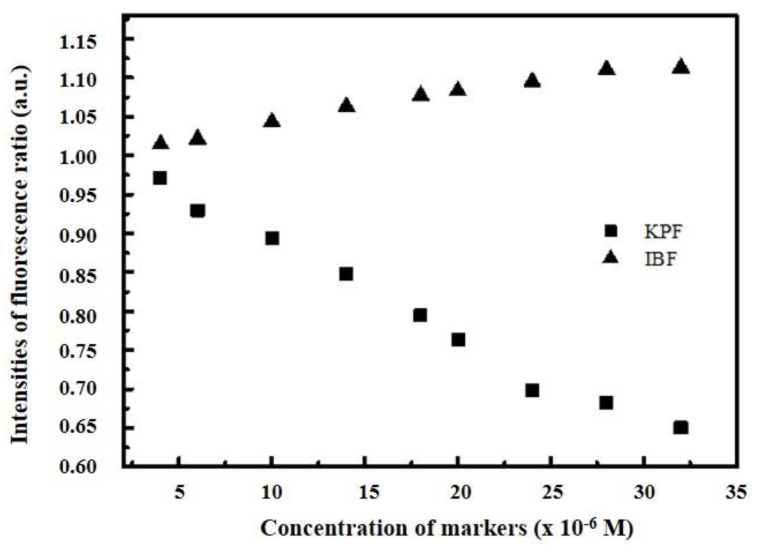
Intensities of fluorescence ratio for prothioconazole/ bovine serum albumin (PTC/BSA) complex in the presence of ketoprofen (KPF) and ibuprofen (IBF) markers.

**Figure 9 ijms-22-01925-f009:**
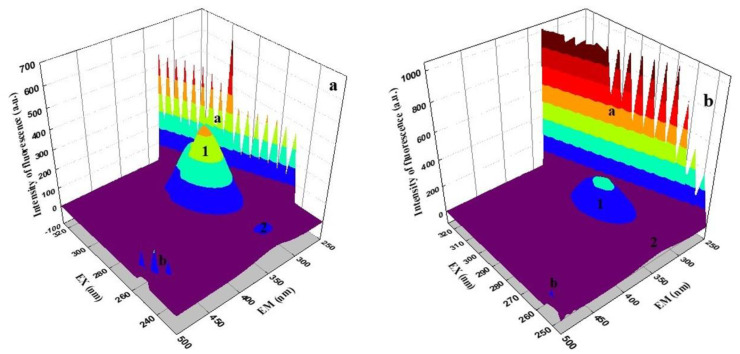
3D fluorescence spectra of bovine serum albumin (BSA) (**a**) and prothioconazole/bovine serum albumin (PTC/BSA) 16/1 (**b**) complex.

**Figure 10 ijms-22-01925-f010:**
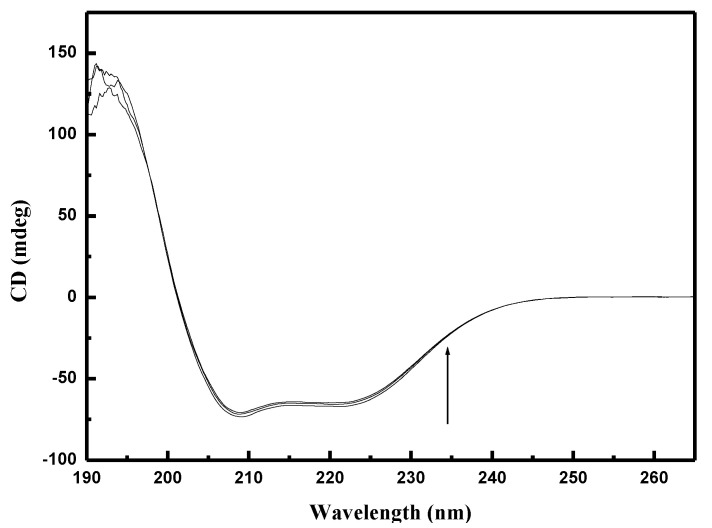
CD spectra of bovine serum albumin (BSA) in the presence of prothioconazole (PTC).

**Figure 11 ijms-22-01925-f011:**
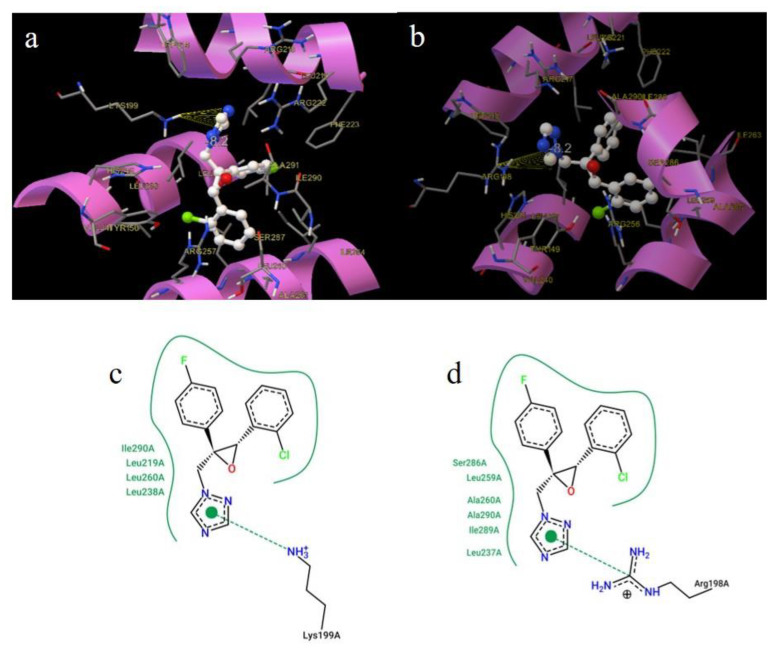
3D reperesentations of epoxiconazole (EPX) in human serum albumin (HSA) (**a**) and bovine serum albumin (BSA) (**b**) Sudlow site I. 2D representations of EPX in HSA (**c**) and BSA (**d**) Sudlow site I.

**Figure 12 ijms-22-01925-f012:**
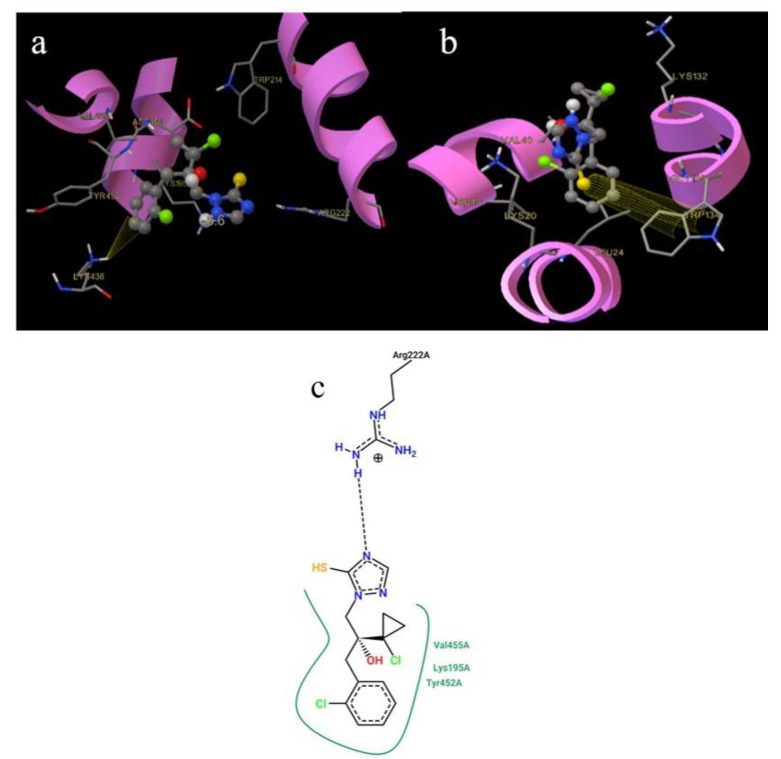
3D representations of prothioconazole (PTC) in human serum albumin (HSA) (**a**) and bovine serum albumin (BSA) (**b**) Sudlow site I. 2D representation of PTC in HSA (**c**).

**Figure 13 ijms-22-01925-f013:**
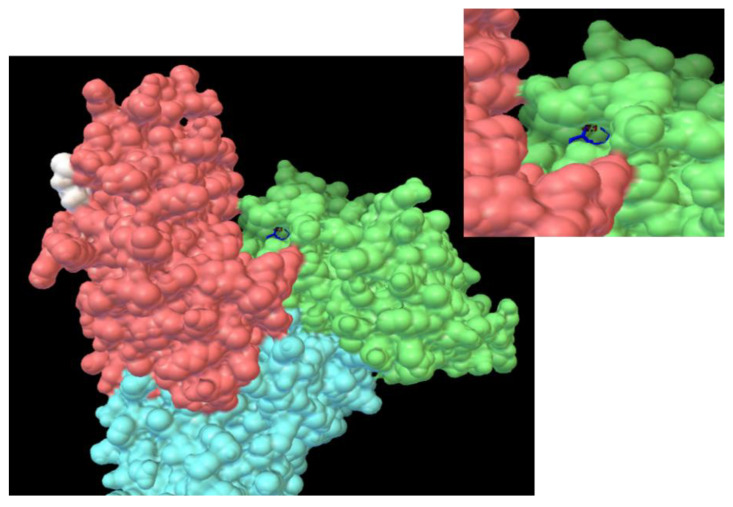
Binding site on domain split for prothioconazole/human serum albumin (PTC/HSA); prothioconazole/bovine serum albumin (PTC/BSA) interaction.

**Table 1 ijms-22-01925-t001:** Binding and thermodynamic parameters for epoxiconazole/human serum albumin (EPX/HSA); epoxiconazole/bovine serum albumin (EPX/BSA) and prothioconazole/human serum albumin (PTC/HSA); prothioconazole/bovine serum albumin (PTC/BSA) interactions.

Complex	T (K)	*K_A_* (L/mol)	*n*	∆*G* (kJ/mol)	∆*H* (kJ/mol)	∆*S* (J/mol.K)
EPX/HSA	298	6.22 × 10^4^	0.98	−26.51	−105.74	−265.88
	303	2.04 × 10^4^	0.95	−25.18		
	310	8.14 × 10^3^	0.91	−23.32		
EPX/BSA	298	3.80 × 10^4^	0.90	−22.14	−81.54	−199.35
	303	1.04 × 10^4^	0.96	−21.13		
	310	7.90 × 10^3^	0.95	−19.73		
PTC/HSA	298	5.75 × 10^5^	1.11	−32.95	−64.39	−105.50
	303	3.80 × 10^5^	1.07	−32.42		
	310	2.08 × 10^5^	0.98	−31.68		
PTC/BSA	298	6.45 × 10^5^	1.06	−32.94	−84.85	−174.18
	303	2.89 × 10^5^	0.99	−32.07		
	310	1.66 × 10^5^	0.95	−30.85		

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
