# Peer review of "Interaction of Conazole Pesticides Epoxiconazole and Prothioconazole with Human and Bovine Serum Albumin Studied Using Spectroscopic Methods and Molecular Modeling"

_ijms, 2021, doi:10.3390/ijms22041925_

Round 1

Reviewer 1 Report

Golianova et al. have investigated interactions between human and bovine serum albumins with two compounds commonly used as pesticides by means of several spectroscopic techniques accompanied by the docking simulations. These pesticides may be potentially hazardous for animals and humans as they were shown to inhibit a number of enzymes related to the hormone biosynthesis. Since serum albumins are responsible for the distribution of chemical compounds throughout the body, it is important to understand their binding poses to them.

The authors measured the binding affinities for the compounds as well as estimated thermodynamic parameters of binding. They also hypothesized several distinct binding sites for the studied pesticides, which can be useful for further characterization of complexes formed by potentially toxic small molecular weight compounds and serum albumin.

The reviewer has several major comments, which should be addressed prior to publication:

  1. In different parts of the manuscript, the authors refer to various BSA/HSA domains and binding sites as well as the number of aromatic amino acids in them (lines 73-82; lines 161-166; line 311; line 325; line 418). Particularly, it does not appear straightforward for the reviewer, how are Sudlow sites I and II, and the warfarin site related? It would be highly beneficial to provide a comprehensive overview of all previously characterized binding sites in BSA/HSA in the introduction including the number/types of aromatic amino acids in them, location in specific protein domains, etc. Also, an additional figure showing these sites will be very helpful, particularly for the interpretation of the docking results.
  2. While discussing the binding parameters (section 2.2, Table 1) and interpreting the Hill plots, the authors refer to the Hill coefficients (n) as a number of independent binding sites. Moreover, based on this interpretation of the Hill coefficients, they make a conclusion about an additional binding site for PTC in HSA. That does not seem correct to the reviewer, as the Hill coefficient describes the degree of cooperativity between ligand binding to cooperative (not independent) binding sites. n > 1 implies positive cooperativity; n < 1 – negative cooperativity; n = 1 – no cooperativity in the ligand binding.
  3. Based on their UV/Vis spectra and CD data, the authors conclude, that the ligand binding induces structural changes in proteins. Given this fact, how can the authors justify the applicability of the rigid docking simulations, which do not consider alterations of protein structure at all?
  4. In their interpretation of CD data, the authors rely on several simple relations between the specific band values and the secondary structure content. To date, there appeared a number of approaches that have been shown to predict the secondary structure content from CD spectra in an accurate way (just a few examples are http://cbdm-01.zdv.uni-mainz.de/~andrade/k2d3/ and http://bestsel.elte.hu/index.php). Perhaps, it would make sense to use one of them/a consensus prediction by several approaches?
  5. Binding poses (in Figures 10, 11) should be accompanied by the 2D diagrams to demonstrate the specific interactions stabilizing the ligands clearly. The Poseview web-server (https://www.zbh.uni-hamburg.de/en/forschung/amd/server/poseview.html) is among possible free solutions allowing to plot such diagrams.
  6. In the Results (line 308-311), the authors state that they used the warfarin site as a center of the docking grid box with the dimensions of 30 x 30 x 30 A. However, in the Methods, they say that the blind docking was carried out prior to it in order to characterize all possible binding poses (How many potential sites were identified? Do they overlap with the sites known from literature?). In the reviewer’s opinion, these results should be at least briefly described in the corresponding section. Also, the reasoning for choosing only the warfarin site for further docking simulations is missing currently. The authors mention in the Methods (line 418) that they “included other binding sites” into their investigation but it appears contradictory to what they state in the Results (see above).

There are a few minor points as well:

  1. Line 72: the term sequence homology is not clear. Please, indicate sequence similarity and sequence identity between BSA and HSA.
  2. Line 77: remove “17-tyrosil” -> “17 tyrosyl”
  3. Line 93: remove either “by” or “using”
  4. Line 142: “the quenching curves decrease” -> “the slope of the quenching curves”?
  5. Line 187: please, use the uniform nomenclature for amino acids, either full names or abbreviations.
  6. In Table 1: please, use dot instead of comma to separate the decimal part.
  7. Line 255: the formulation “binding of PTC to the surface of the BSA molecule” sounds very uncertain. Is it possible to propose at least a hypothetical site?

Author Response

Responses to Reviewer  Comments

Point 1: In different parts of the manuscript, the authors refer to various BSA/HSA domains and binding sites as well as the number of aromatic amino acids in them (lines 73-82; lines 161-166; line 311; line 325; line 418). Particularly, it does not appear straightforward for the reviewer, how are Sudlow sites I and II, and the warfarin site related? It would be highly beneficial to provide a comprehensive overview of all previously characterized binding sites in BSA/HSA in the introduction including the number/types of aromatic amino acids in them, location in specific protein domains, etc. Also, an additional figure showing these sites will be very helpful, particularly for the interpretation of the docking results.

Response 1: We have inserted a following part concerning with more detailed description of binding sites within protein domains into Introduction. Characterization of binding sites is given separately for HSA and BSA respectively and associated with Figure 2 a,b. 

"HSA has extraordinary binding properties. It can bind under the physiological conditions not only fatty acids, peptides and proteins, but also low molecular weight endogenous and exogenous molecules [23, 24]. HSA binds medium and long saturated fatty acid molecules (C10 – C18) and long chain unsaturated fatty acid molecules (arachidonic and oleic acid) up to seven binding site (FA1 – FA7). They are located inside subdomains (FA1, FA4, FA5 and FA7), at the interface of domains (FA2 and FA3) and at the interface of two subdomains belonging to one domain (FA6) (Figure 2a). HSA has a high affinity site for binding metals to N-ends of the chain [11], where bind mainly Zn(II), Cd(II), Cu(II) and Ni(II) [25]. Domains II and III contain two primary drug binding sites, known as Sudlow´s site I and II (Figure 2a) [26]. Sudlow´s site I is the primary binding site for warfarin, often called warfarin binding sites instead, but molecules such as phenylbutazone and azapropazone also bind there. It is located in cavity of subdomain IIA, which is for the most part nonpolar. An important part of Sudlow´s site I is the only on Trp present in HSA (Trp214), as its indole ring and its rotation allow by a certain ligand nesting into the Sudlow´s site and creating the stacking interactions. Molecules that specifically bind into the Sudlow´s site I are bulky negatively charged heterocyclic molecules located in the middle of them. Sudlow´s site II is the primary binding for ibuprofen and diazepam. Compared to Sudlow´s site I, it is smaller and contains only one binding region. While Sudlow´s site I is surrounded by subdomains IIB and IIIA, Sudlow´s site II is surrounded by subdomains IIIB and IIIA, which different rotation provides direct contact of Sudlow´s site II with the solvent. Sudlow´s site II binding molecules are predominantly aromatic carboxylic acids which have negative charge on the alpha carbon [27-29].

Figure 2 can be found in manuscript

BSA has several binding sites, of which between the most important are sites I and II located in the hydrophobic cavities of subdomains IIA and IIIA [31]. Markers of these binding sites include warfarin, phenylbutazone, dansylamide, iodipamide (binding site I) and ibuprofen, flufenamic acid and diazepam (binding site II) (Figure 2b). Unique binding properties are manifested in binding with nucleic acids, proteins, coordination compounds [32-34]. BSA also has the ability to bind large amounts of ions (Cu2+, Ni2+, Zn2+, Co2+, Pt2+) [35-37]. Some degree of binding activity was also observed for metal complexes due to weak binding interactions between metals and the tryptophan residue Trp134, which is located on the surface (it is more available than Trp213) [38].  "  

Point 2: While discussing the binding parameters (section 2.2, Table 1) and interpreting the Hill plots, the authors refer to the Hill coefficients (n) as a number of independent binding sites. Moreover, based on this interpretation of the Hill coefficients, they make a conclusion about an additional binding site for PTC in HSA. That does not seem correct to the reviewer, as the Hill coefficient describes the degree of cooperativity between ligand binding to cooperative (not independent) binding sites. n > 1 implies positive cooperativity; n < 1 – negative cooperativity; n = 1 – no cooperativity in the ligand binding.

Response 2: Yes, you right, the Hill coefficient is about cooperativity and we have input short correction into the text: we have substituted independent by "additional" (row 186); we have rewrote the sentence in row 200 into: "additional binding site given by cooperation with regular binding site" (row 200); added "additional" (row 202).

Point 3: Based on their UV/Vis spectra and CD data, the authors conclude, that the ligand binding induces structural changes in proteins. Given this fact, how can the authors justify the applicability of the rigid docking simulations, which do not consider alterations of protein structure at all?

Response 3: The justification is that the simulation results are valid for a small range of protein conformation change as the computations themselves are approximative. This leavey  is used readily in other literature. We have further added the description of this limitation. We also have the benefit of experimental evidence that helps us pair the simulated results to empirical measurements so that we can easily eliminate any gross simulation artefacts.

Point 4: In their interpretation of CD data, the authors rely on several simple relations between the specific band values and the secondary structure content. To date, there appeared a number of approaches that have been shown to predict the secondary structure content from CD spectra in an accurate way (just a few examples are http://cbdm-01.zdv.uni-mainz.de/~andrade/k2d3/ and http://bestsel.elte.hu/index.php). Perhaps, it would make sense to use one of them/a consensus prediction by several approaches?

Response 4: Thank you for recommendation - it is very valuable for us. We used the softwer you proposed and calculated the secundary structure content. Following text is including in manuscript (rows 348-353):

"We used the web server K2D3 [63] to verify the content of α-helix and calculate β-strand of the protein from its circular dichroism spectrum. Using this program, we found that free HSA and BSA contained 12.69 % and 11.69 % β-strand respectively. By adding EPX and PTC to the solution, we noticed a gradual increase in β-structures. The increasing of β-strands was more modest than the decrease in α-helixes."

Point 5: Binding poses (in Figures 10, 11,) should be accompanied by the 2D diagrams to demonstrate the specific interactions stabilizing the ligands clearly. The Poseview web-server (https://www.zbh.uni-hamburg.de/en/forschung/amd/server/poseview.html) is among possible free solutions allowing to plot such diagrams.

Response 5: We have added the 2D representations into Figures 10,11. This is a suboptimal solution as we could not find any software capable of displaying the unexpected π-π interaction of the PTC with BSA-TRP134. And to keep the visual consistent we kept the 3D images in the main paper body.

Point 6: In the Results (line 308-311), the authors state that they used the warfarin site as a center of the docking grid box with the dimensions of 30 x 30 x 30 A. However, in the Methods, they say that the blind docking was carried out prior to it in order to characterize all possible binding poses (How many potential sites were identified? Do they overlap with the sites known from literature?). In the reviewer’s opinion, these results should be at least briefly described in the corresponding section. Also, the reasoning for choosing only the warfarin site for further docking simulations is missing currently. The authors mention in the Methods (line 418) that they “included other binding sites” into their investigation but it appears contradictory to what they state in the Results (see above).

Response 6: Based on your input we included a short description of the other notable binding site, specifically the Ibuprofene binding site (Sudlow site II). The reasoning for no further investigation into the other binding sites is significant lowering of binding energy. This information has been added to the text. Additionally the resulting binding of BSA-PTC is a result of the blind binding.

Minor comments:

Point 1: Line 72: the term sequence homology is not clear. Please, indicate sequence similarity and sequence identity between BSA and HSA 

Response 1: We have substituted "homology" by "similarities" (row 72) and inserted some differences between BSA and HSA on rows 76-80

Point 2: Line 77: remove “17-tyrosil” -> “17 tyrosyl”

Response 2: We have corrected it.

Point 3: Line 93: remove either “by” or “using”

Response 3:  We have removed "by"

Point 4: Line 142: “the quenching curves decrease” -> “the slope of the quenching curves”?

Response 4: We substituted "the quenching curves decrease” by “the slope of the quenching curves”

Point 5: Line 187: please, use the uniform nomenclature for amino acids, either full names or abbreviations.

Response 5: We did it.

Point 6: In Table 1: please, use dot instead of comma to separate the decimal part.

Response 6: We did it.

Point 7: Line 255: the formulation “binding of PTC to the surface of the BSA molecule” sounds very uncertain. Is it possible to propose at least a hypothetical site?

Response 7: We have added a sentence: "As it will be shown below in the text, our further results confirm this proposition and this binding site is located on domain split in the albumin molecule (Figure 13)." for better specification of binding site.

Let me thank you for your inspiring questions and comments - we have learned a lot and obtained a new view for our results.

Reviewer 2 Report

In this manuscript, the authors systematically investigated the binding effect of conazole pesticides, EPX and PTC, on human and bovine serum albumin (HAS and BSA) by ground state electronic spectroscopy and molecular docking. The fluorescence and UV/Vis results reveal the binding affinity, local molecular interaction and conformational change. Along with the molecular modeling, specific binding sites are discovered. This work is a significant contribution to the field and should be published on IJMS. I only have a few questions that need clarification.

1. Figure 5. The authors said that “No changes in the behavior of peak 2 were observed”. However, it is clear that the intensity of peak 2 is decreased by adding more concentrated EPX and PTC. What is the reason?

2. Figure 8, it would be better if the authors can show the two figures (a and b) with the same perspective and switch the EM and EX axis side. So the all the peaks in the emission spectra can be clearly seen. And it would also be helpful to label all the peaks (a,b,1 and 2) on the figure.

3. No shifts in emission wavelength were observed on Figure 8, why does it suggest a conformational change of BSA and HAS? 

Author Response

Responses to Reviewer Comments 

Point 1: Figure 5. The authors said that “No changes in the behavior of peak 2 were observed”. However, it is clear that the intensity of peak 2 is decreased by adding more concentrated EPX and PTC. What is the reason?

Response 1:  Yes, you wright, the last sentence in paragraph 2.4 can be confusing. This sentence follows our previous discussion about wavelength shift of peak 1 (not intensity). We have intended to postulate that the changes in intensity and wavelength of peak 2 are considerably small in comparison with those of peak 1. The sentence was changed as follows: "No changes in the behavior wavelength shift of peak 2 were observed." 

Point 2: Figure 8, it would be better if the authors can show the two figures (a and b) with the same perspective and switch the EM and EX axis side. So the all the peaks in the emission spectra can be clearly seen. And it would also be helpful to label all the peaks (a,b,1 and 2) on the figure.

Response 2: Figure 8 has been completely remade with respect to your proposed recommendations. The perspective was unified, side axis were canceled and labels of all the peaks were added. At the same time, Figures S13 -S15 in Supplementary Material have been remade, too. In the new version of manuscript, the number of Figure 8 has changed to Figure 9 and the numbers of Figures S13 - S15 have changed to S14 - S16.

Point 3: No shifts in emission wavelength were observed on Figure 8, why does it suggest a conformational change of BSA and HAS? 

Response 3:  Slight conformational change of SA after adding pesticide molecules can be proposed by following explanation: "Peak 1 represents the fluorescence contribution of Tyr and Trp and peak 2 represents the spectral properties of the polypeptide structural chain, its intensity correlated with the secondary structure of the protein [58]. We can observe a decreasing tendency of peak 1 after adding individual conazoles to SA solutions (Figure 9). This decrease indicates the binding of conazoles near the Tyr or Trp of the individual proteins, which is consistent with fluorescence measurements. We can also observe a decrease in peak 2, which indicates changes in the peptide structure. Its declining trend after conazoles binding suggests that there is a slight destabilization of individual proteins and a slight unfolding of the polypeptide chain leading to conformational changes [58]. This result correlates with absorption and CD spectroscopy measurements." We included this additional text into paragraph 2.7.

Let me thank you for your inspiring questions and comments - we have learned a lot and obtained a new view for our results.

Reviewer 3 Report

The manuscript by Golianova et al. investigated the binding of conazole fungicides with 423 serum albumins using spectroscopic methods complemented with molecular modeling. 

The modeling seems fine and is according to experimental results. Publish as is.

Author Response

We are thankful for your revision. We appreciate your positive review very much.

Round 2

Reviewer 1 Report

The paper has been significantly improved and it can be now published in IJMS.